# The Transcriptomic Profiles of *ESR1* and *MMP3* Stratify the Risk of Biochemical Recurrence in Primary Prostate Cancer beyond Clinical Features

**DOI:** 10.3390/ijms24098399

**Published:** 2023-05-07

**Authors:** Michał Olczak, Magdalena Julita Orzechowska, Andrzej K. Bednarek, Marek Lipiński

**Affiliations:** 1II Clinic of Urology, Medical University of Lodz, Pabianicka 62, 93-513 Lodz, Poland; 2Department of Molecular Carcinogenesis, Medical University of Lodz, Żeligowskiego 7/9, 90-752 Lodz, Poland

**Keywords:** prostate cancer, prostate-specific antigen, biochemical recurrence, disease progression, high-risk, estrogen receptor, androgen receptor, epithelial-to-mesenchymal transition

## Abstract

The molecular determinants of the heterogenic course of prostate cancer (PC) remain elusive. We aimed to determine the drivers predisposing to unfavorable PC outcomes anticipated by BCR events among patients of similar preoperative characteristics. The TCGA transcriptomic and clinical data of 497 PC individuals were used, stratified according to the risk of BCR by EAU-EANM-ESTRO-ESUR-SIOG. The relevance of the functional markers regarding BCR-free survival was examined by the cutp algorithm. Through UpSetR, subgroups of PC patients bearing an unfavorable signature were identified, followed by the hierarchical clustering of the major markers of the epithelial-to-mesenchymal transition (EMT). BCR-free survival was estimated with the Cox proportional hazards regression model. *ESR1* significantly differentiated BCR-free survival, whereas *AR* did not. An elevation in *KLK3* correlated with better prognosis, although *PGR*, *KLK3*, *CDH1*, and *MMP3* predicted BCR better than the preoperative PSA level. Patients sharing an unfavorable profile of *ESR1* and *MMP3* together with lymph node status, Gleason score, T, and EAU risk groups were at a higher risk of BCR originating from mesenchymal features of PC cells. To conclude, we revealed an *ESR1*-driven unfavorable profile of EMT underpinning a worse PC trajectory. *ESR1* may have a major role in PC progression; therefore, it could become a major focus for further investigations.

## 1. Introduction

Prostate cancer (PC) is the second most common cancer and the fifth most common solid tumor to cause death in men worldwide, with an estimated 1,400,000 new cases annually and a total of over 375,000 deaths each year. Its incidence and mortality rates vary between regions, being the highest in Northern Europe and Central America [1]. In addition to its high prevalence, PC-related prognosis diverges with patients’ age [2], ethnicity, stage [3], and genetic background. The above, together with inherited susceptibility, obesity, and environmental influence, are also recognized as major risk factors for the disease [4]. Currently, several germline variants of i.e., *HOXB13*, *NKX3.1*, and *BRCA2* are associated with an increased risk of inherited susceptibility to PC. These mutations are more common in men with early-onset PC arising due to congenital defects in response to oxidative stress (e.g., *NKX3.1*) or DNA damage repair mechanisms (*BRCA2*); therefore, they may predict a more aggressive course of the disease. On the other hand, these variants are of relatively low frequency among all PC cases and do not resemble the etiology of sporadic PC [5,6,7].

PCs are considered a flagship example of the intra-tumoral heterogeneity standing behind the inequities in the course of the disease observed between patients. Such complexities arise from the multifocal nature of the disease currently often associated with the clonal branching evolution of different molecular architectures and genomic burdens that ultimately poses a barrier to therapeutic efficacy [8,9,10]. It may be inferred that the loading of the genomic alterations, both germline and somatic, has a predominant impact on the development and progression of the tumor [11]. In fact, these are the actual markers driving the individual trajectory of PC, whose prediction, contrary to general opinion, lies beyond the routine histopathological features.

Radical prostatectomy (RP) is one of the primary treatment options for localized PC, and it has excellent oncologic outcomes. Despite the good overall control of the disease progression after RP, 20–30% of patients experience PC recurrence that manifests in an increase in the prostate-specific antigen (PSA) in the serum without any clinical evidence of metastatic disease (MD) [12]. A PSA elevation is thereby an indicator of remaining prostatic tissue, denoting the presence of PC cells at local or distant sites. Although biochemical recurrence (BCR) anticipates PC progression, the increase in the PSA concentration may be prolonged and does not always lead to clinically apparent metastases [13]; nevertheless, it has been reported that the 5-year risk of clinical progression (CP) in men with BCR ranges from 27% to 60% [12].

In the current recommendations, the European Association of Urology (EAU) proposes risk group classification combining the Gleason grading score (GGS), clinical stage (TNM), and initial PSA level [14]. This classification derived primarily from the study by D’Amico et al. [15], which was based on the stratification of patients with a similar risk of BCR at the time of primary intervention (RP or external beam radiotherapy (EBRT)) and is commonly used in clinical settings [16].

The clinical use of PSA as a routine marker of PC is now up for debate due to many controversies. In particular, among symptomatic patients, the specificity of PSA was found to be as low as 3% at 96% sensitivity; however, these estimations varied between different studies and applied only to the PSA threshold of 4 ng/mL in PC detection [17]. This was explained by the fact that PSA is not tumor-specific, but rather an organ-specific marker that is entirely produced by the prostatic gland [18]. It is also a prostate differentiation marker; thus, tumors with higher Gleason scores that have a more aggressive clinical course tend to express lower values of PSA [19]. Moreover, the contribution of the tumor alone to the elevated PSA value is unpredictable, as it may also be derived from an enlarged prostatic gland with hyperplasia, prostate inflammation, or manipulation by a prostate massage or biopsy. Additionally, studies show that the range of PSA values increases with age, and so this marker should be interpreted with caution and considered collectively with other clinical parameters [20]. Therefore, the use of PSA as a screening tool involves the risk of overdiagnosis and overtreatment, causing patients with insignificant cancer to be subjected to unnecessary treatment for an indolent disease. Nevertheless, the standard PSA threshold of 4 ng/mL, determined to maximize cancer detection, is currently considered too low and causes the misidentification of low-grade or benign cases [21]. More importantly, there is no consensus regarding the levels of PSA defining BCR following RP. The values suggested in the current literature range between 0.2 and 0.6 ng/mL, with values above the cutoff of 0.4 ng/mL best predicting further progression. The value of PSA after RP is not expected to exceed 0.1 ng/mL, while values above 0.1 ng/mL are recognized as the most indicative of residual prostate tumor tissue [22,23]. Regardless, PSA is widely used in many nomograms (e.g., EAU risk groups) to estimate the risk of BCR, but also as a part of various assessment tools by Boorjian [24], Briganti [25], and Partin [26] to evaluate the risk of CP and cancer-specific and overall survival (CSS and OS, respectively) rates after an RP of acceptable accuracy.

Apart from PC cases that meet the EAU recommendations for PSA-based prognostication, of the utmost importance should be distinguishing aggressive from indolent tumors of high recurrence potential. Recently, studies have described specific subgroups of patients classified with a low GGS, PSA level, and/or disease risk who presented unusually aggressive courses of PC, as observed in a clinical routine [27]. All of the above prompted us to investigate the molecular mechanisms underpinning distinct trajectories of PC progression that could be adopted in clinical care and better stratify patients with rapid recurrence in spite of the favorable primary prognosis.

The epithelial-to-mesenchymal transition (EMT) is an essential process during cancer progression enabling tumor cells to invade local tissues, disseminate, and ultimately form distant metastases that naturally occur during tissue remodeling or embryogenesis. Specifically, this cellular mechanism implies the loss of function and structure of epithelial cells that, at the end of the transition, present a mesenchymal phenotype and therefore acquire migratory and invasive properties. The EMT is regulated by various transcription factor (TF) families, such as Zeb (Zeb1, Zeb2); Snai (Snai1, Snai2); and Twist, switching the expression profiles of specific genes determining either the epithelial (e.g., *CDH1*) and/or mesenchymal state (e.g., *FN1*, *CDH2*, *VIM*). As a result, it involves the remodeling of the cytoskeleton and local tissue architecture, the loss of cell–cell junctions, changes in cell polarity, and the destabilization of interactions with the extracellular matrix (ECM), ultimately allowing the cells to escape from the primary tumor site. Once the new foci in a distant tissue have been formed, the cells regain their epithelial character through a reverse program—a mesenchymal-to-epithelial transition (MET), enabling the colonization of a new niche. Many growth factors and signaling pathways orchestrate the EMT, dictating thus the nature of neoplastic cells and the trajectory of cancer progression [28,29]. Notably, it has been shown that the specific profiles of the EMT reflect the clinically observed Gleason score delineating the levels of cellular de-differentiation within the PC foci [2], likely as a transcriptional response to either androgen or estrogen signaling axes [30].

In previous studies, other important factors affecting the mechanism of PC recurrence and progression (i.e., EMT) were shown, such as age and signaling by steroid hormone receptors (*AR*, *ESR1*, *ESR2*), revolving around the remodeling of the tumor tissue architecture and encouraging more aggressive clinical behavior from PC cells, consequently tipping the odds to favor the establishment of overt PC progression among specific subgroups of patients [2,3]. However, these studies did not go beyond the Gleason score to routinely classified subgroups of patients according to, e.g., the EAU guidelines and did not correlate the molecular profiles to any specific prognosis. Nevertheless, BCR is considered a primary sign of PC progression and a more aggressive course of the disease, although its accurate diagnosis utilizing routine PSA levels in the serum is generally perplexing and may lead to poor patient outcomes. Altogether, the above prompted us to determine the association of BCR with its molecular drivers, i.e., to identify functional biomarkers of pivotal relevance in EMT with regards to clinical parameters such as PSA, GGS, and T score to assess the risk of BCR at the time of primary treatment and enable the stratification of the potential for a more aggressive course of PC. We designed the study to be a priori oriented towards the genes relevant to the EMT to present a holistic view of the molecular aspects of PC progression using a set of functional analyses, which resulted from the molecular function of the genes and gene sets explaining the observed phenotypes and outcomes.

## 2. Results

### 2.1. Transcriptomic Markers Improve Preoperative Prediction of BCR among PRAD Patients

Previously, we observed the disparate significance of hormone receptors such as *AR*, *ESR1*, and *ESR2* regarding PRAD recurrence, which changed with increasing age [2]. In the present study, we thus evaluated the preoperative prognostic ability of hormonal receptor profiling (*AR*, *ESR1*, *ESR2*, and *PGR*) accompanied by the other functional markers that play a pivotal role in the mechanism of tumor recurrence (*CDH1*, *VIM*, *MMP2*, *MMP3*, *MMP9*, and *CTNNB1*) and PRAD management (*KLK2*, *KLK3*, *FOLH1*, *NR3C1*, and preoperative PSA serum level) with BCR by applying the cutpoint optimization algorithm. This algorithm determines the cutpoint of the most significant split in the Kaplan–Meier estimator stratifying patients into subgroups of favorable/unfavorable BCR-related prognosis based on the expression of a particular gene. During the overall follow-up time of 4604 days for BCR-free survival among the PRAD cohort, 72 patients (14%) experienced BCR events, with 2- and 5-year BCR-free rates of 84% (95% CI: 80–88%) and 64% (95% CI: 56–74%), respectively (Appendix A).

Regarding the hormonal receptors, *ESR1* and *PGR* significantly differentiated the BCR-free survival of PRAD patients, but surprisingly, *AR* did not (Table 1). In particular, a lowered expression of *ESR1* correlated with a more favorable prognosis (HR = 1.87, 95% CI = 1.06–3.31, *p* = 0.029), with 2- and 5-year BCR-free rates reaching 88% and 82% in comparison with 78% and 57% for the unfavorable group, respectively (Figure 1, Table 1). In contrast, a lowered expression of *PGR* was associated with more frequent BCR among patients (HR = 0.466, 95% CI = 0.267–0.813, *p* = 0.006), showing 2- and 5-year BCR-free rates of 68% and 43% vs. 86% and 67%, respectively, for patients with a higher expression of *PGR* (Figure 1, Table 1). 

*KLK3* encoding PSA and *FOLH1* encoding PSMA showed adversative profiles correlating with better clinical outcomes. Specifically, a higher expression of *KLK3* was more favorable (HR = 0.423, 95% CI = 0.265–0.677, *p* = 0.0002; 2-year BCR-free rate: 88% vs. 73%; 5-year BCR-free rate: 69% vs. 50%), whereas lowered *FOLH1* correlated with a higher BCR-free rate (HR = 1.6, 95% CI = 0.986–2.58, *p* = 0.054; 2-year BCR-free rate: 85% vs. 80%; 5-year BCR-free rate: 70% vs. 51%) (Figure 1, Table 1). In turn, an increased expression of *CDH1*, encoding the epithelial-state marker E-cadherin, correlated with an improved BCR prognosis (HR = 0.545, 95% CI = 0.342–0.868, *p* = 0.009; 2-year BCR-free rate: 88% vs. 76%; 5-year BCR-free rate: 70% vs. 53%), in contrast to *VIM*, encoding the mesenchymal-state marker vimentin, whose decreased expression was associated with a more favorable outcome (HR = 1.85, 95% CI = 1.11–3.07, *p* = 0.016; 2-year BCR-free rate: 85% vs. 80%; 5-year BCR-free rate: 68% vs. 51%). Similarly, a lowered expression of *MMP3* was associated with a more favorable prognosis (HR = 2.93, 95% CI = 1.18–7.27, *p* = 0.015), demonstrating BCR-free rates reaching 92% vs. 82% for 2-year follow-up and 84% vs. 60% for 5-year follow-up (Figure 1, Table 1). Finally, we established a cutpoint for the preoperative PSA concentration of 6.9 ng/mL, which significantly differentiated BCR-free survival among the PRAD cohort (HR = 1.93, 95% CI = 1.15–3.25, *p* = 0.012). Specifically, PSA values below 6.9 ng/mL correlated with better outcomes, demonstrating 90% vs. 80% 2-year BCR-free rates and 70% vs. 60% 5-year BCR-free rates (Figure 1, Table 1).

### 2.2. The Specific Combination of ESR1 and MMP3 Discriminates EMT-Related Potential for PRAD Recurrence

We further attempted to identify subgroups of PRAD patients who bore a specific profile (i.e., overlapping unfavorable expression profiles of particular predictors significantly differentiating BCR-free survival) corresponding to worse clinical outcomes that increased the molecular potential for PRAD recurrence. The UpSet analysis revealed a subgroup of 296 patients (59.5%) who shared a common profile of unfavorable BCR prognosis regarding the expression of *ESR1* and *MMP3* (Appendix A). The resultant signature, hereinafter called the “stratum”, comprised the simultaneous heightening of the *ESR1* and *MMP3* expression above the determined cutpoints (Table 1).

Notably, we observed a remarkable difference in BCR-free survival rates between patients stratified into the identified stratum (HR = 2.2, 95% CI = 1.3–3.6, *p* = 0.003), especially in the 5-year follow-up period, reaching 78% among non-carriers vs. 54% in carriers of the signature (Figure 2, Table 2). We combined the major clinical parameters with the stratum to compare the differences in the course of PRAD between clinically equivalent cases, i.e., whether (e.g.,) a Gleason 6 patient with an identified molecular profile would differ in BCR prognosis from a Gleason 6 patient without that molecular profile. We observed significant differences regarding the stratum and lymph node status (global log-rank *p* = 0.0015), Gleason score (global log-rank *p* < 0.0001), TNM (global log-rank *p* = 0.027), and stage EAU risk groups (global log-rank *p* < 0.0001) (Figure 2). In the univariate model of Cox proportional hazards, the presence of the stratum increased the risk of BCR, especially in the absence of positive lymph nodes (stratum-negative pN0 vs. stratum-positive pN0 HR = 2.58, 95% CI = 1.27–5.2, *p* = 0.009), thus decreasing the 5-year BCR-free rate from 81% to 55%. As expected, the positive status of the lymph nodes was considered a predictor of worse outcomes, which in the present study was additionally modified by the presence of the stratum (stratum-negative pN0 vs. stratum-negative pN1 HR = 3.17, 95% CI = 1.2–8.3, *p* = 0.02, vs. stratum-positive pN1 HR = 4.65, 95% CI = 2.02–10.7, *p* < 0.001). The model including the Gleason score combined with the stratum showed no significant differences in the BCR-free survival of stratum-negative Gleason 7A (3 + 4) vs. stratum-positive Gleason 6 or vs. stratum-positive Gleason 7A (3 + 4), although this could have been limited by the low number of events in these groups. Nevertheless, we observed a striking corresponding decrease in the restricted mean survival times: 4244.59 (SE_rmean_ = 269.05) for stratum-negative Gleason 7A (3 + 4), 2336.27 (SE_rmean_ = 186.78) for stratum-positive Gleason 6, and 2863.35 (SE_rmean_ = 395.79) for stratum-positive Gleason 7A (3 + 4). Notably, higher Gleason scores accompanied by the presence of the stratum strongly modified the BCR prognosis. The worst outcomes were reported for stratum-positive Gleason 9 (HR = 13.7, 95% CI = 3.27–57.4, *p* < 0.001 vs. stratum-negative Gleason 7A (3 + 4)), showing a 36% 5-year BCR-free survival probability in comparison with the 62% probability among stratum-negative Gleason 9 patients. Moreover, patients categorized according to the EAU risk groups as cT2c-4 bearing the stratum showed twice the risk of BCR in comparison with non-carriers of the signature (HR = 2.05, 95% CI = 1.21–3.46, *p* = 0.008), with a remarkable decrease in the 5-year BCR-free survival rate from 75% to 51%. Although the results of the univariate model including the stage of PRAD were equivocal, there was a consistent trend towards a worse BCR-free survival rate among localized stratum-negative vs. stratum-positive PRAD cases and locally advanced disease, respectively (global log-rank *p* < 0.0001). A significant difference in BCR prognosis was also reported for stratum-positive patients with PSA levels of 10–20 ng/mL (HR = 2.23, 95% CI = 1–4.97, *p* = 0.049) and >20 ng/mL (HR = 2.83, 95% CI = 1.16–6.94, *p* = 0.023) in comparison to non-carriers of the signature. Finally, as the most conclusive evidence supporting our hypothesis, we demonstrated striking disparities within the subgroup of patients restricted to only those with a high risk of BCR (according to the EAU guidelines) who were stratum-positive in comparison to high-BCR-risk non-carriers of the signature (HR = 2.1, 95% CI = 1.2–3.67, *p* = 0.01). However, due to the low sample sizes, we failed to perform analogous comparisons for the low- and intermediate-risk groups for BCR. The Kaplan–Meier curves representing the BCR-free survival analyses performed under the aforementioned clinical settings are shown in Figure 2, with detailed statistics summarized in Table 2.

As an additional finding, we observed significant differences in BCR-free rates regarding certain clinical features, independently of the stratum, such as lymph node status (*p* = 0.0046), stage group (*p* < 0.0001), and Gleason score (*p* < 0.0001), in contrast to the preoperative PSA and TNM categories of the EAU, which did not stratify the BCR-free survival of PRAD patients in a significant manner (*p* = 0.72, *p* = 0.095, respectively; Appendix A).

Concerning the disproportions in the sizes of the patient subgroups classified according to the stratum combined with clinical parameters analyzed in terms of BCR-free survival, we additionally employed CA (the PCA-related alternative approach to classical contingency tables) to reveal the topological associations between all these variables among the PRAD cohort that could not be ascertained by the standard approach (descriptive statistics comparing the subgroups with a χ^2^ independence test or Fisher exact test). By revealing the spatial distribution of the clinical features of specific levels along with the combination of the stratum and BCR status, the CA fully described the overall characteristics of the individual subgroups of PC patients with a molecular predisposition to BCR-anticipated progression. As may be seen in Figure 3, the CA revealed that the presence of the stratum in the absence of BCR correlated with a preoperative PSA level of 10–20 ng/mL and a Gleason score of 7A (3 + 4) as well as 7B (4 + 3), whereas the absence of the stratum partitioned the variables along with dim2 and pN0, localized PRAD, and Gleason 6 and 7A (3 + 4). In contrast, the occurrence of BCR regardless of the stratum status was correlated with a preoperative PSA concentration >20 ng/mL, Gleason 8 and 9, pN1, and a locally advanced stage of the disease. Importantly, some of these observations could be confirmed with a cross-validation study (Appendix A).

Next, as we had identified the molecular profile of unfavorable BCR-related prognosis comprising *ESR1* and *MMP3*, we investigated the downstream biological consequences possibly reflecting the increased risk of PRAD recurrence; therefore, we focused on the transcriptional effectors of *ESR1* involved in the EMT (41 genes; see Materials), considered a key mechanism for the relapse of the disease and invasive potential. The heatmap shown in Appendix A represents the opposing profiles of selected EMT markers among stratum-negative vs. stratum-positive PRAD patients. A heightened expression of epithelial-state markers such as *CDH1*, *LAMA1*, and *KRT18* with a simultaneous decrease in the expression of, among others, *CTNNB1*, *FN1*, *MMP2*, *MMP3*, *MMP9*, *CDH2*, *ACTA2*, *SNAI2*, *VIM*, *ZEB1*, and *ZEB2* was observed among the former subgroup, whereas the latter demonstrated the opposite, with mesenchymal-state-associated expression patterns of the aforementioned genes reflecting the worse clinical outcomes.

Subsequently, we differentiated the profiles of the EMT according to the stratum and clinical variables as a molecular indicator of the BCR-free survival background; thereby, we revealed the continuum of the EMT-related states switching between the subgroups, especially observable in the presence/absence of the stratum. Notably, we observed some specific expression patterns of the genes that were repeatable independently of the studied variable. In particular, the expression of epithelial markers such as *CDH1*, *DSP*, *KRT8*, *KRT18*, and *LAMA5* decreased with an advancing disease, indicated by factors such as the presence of the stratum combined with either pN1 or an increasing preoperative PSA level, Gleason component, stage, or TNM group (Figure 4). On the contrary, mesenchymal *ACTA2*, *MMP2*, *MMP3*, *MMP9*, *ITGA5*, *ITGB6*, *CDH2*, *CDH11*, *VIM*, *CTNNB1*, *FN1*, *SMAD2*, *SMAD3*, *SNAI2*, *LEF1*, *TCF3*, *TCF4*, *ZEB1*, and *ZEB2* were heightened among stratum-positive patients, regardless of the clinical phenotype. Notably, an increased expression of *TCF3* and *GSC* was specifically observed among pN1 cases (Figure 4A), locally advanced stages of the disease (Figure 4D), and cT2b patients (Figure 4E), regardless of the presence of the stratum. Importantly, many of these findings were confirmed in the cross-validation analysis employing the independent Rubicz et al. cohort [31] (Figure 5).

## 3. Discussion

PC is the major cancer-related cause of death among males worldwide [1]. The progression of PC is preceded by BCR, manifesting in an increase in serum levels of PSA. It is estimated that approximately 20–30% of patients will experience BCR in the 5 years following treatment, and, more importantly, 24–34% of them will develop PC metastasis with a high burden of lethality [32,33,34]. In our study, during the overall follow-up time of 12.6 years (4604 days) post-RP, 72 patients (14%) experienced BCR events, with a 5-year rate of 64% (95% CI = 56–74%, Appendix A). Our observations were similar to those from the study by Han et al. of 2091 PC patients reporting a BCR rate of 17% during 17 years of follow-up and utilizing the same cutpoint for serum PSA of 0.2 ng/mL to define BCR [35].

Currently, multiple treatment modalities are available for advanced or metastatic castration-resistant PC (mCRPC), although they are largely considered palliative options and prolong the median survival time only to a marginal extent [36,37,38,39]. In clinical practice, the diacritic course of PC is very heterogeneous and often progresses with a high degree of disparity observed between patients of similar characteristics. The paradigm of restricted therapeutic outcomes together with an unclear silhouette of PC in an individual is now beginning to shift research interests towards the potential of an accurate stratification of the recurrence risk, i.e., BCR progression. Interestingly, this phenomenon was confirmed, as an illustrative instance, in the clinical characteristics of the PC cohort analyzed in this study. In general, as reported in the literature, the risk of BCR tends to increase with high levels of PSA and an advancing pathologic stage, GGS, and lymph node status [35]. The EAU-EANM-ESTRO-ESUR-SIOG guidelines adopted a model by D’Amico et al. stratifying BCR prognosis by attributing patients with the above features, which, when combined, defined the risk as either low, intermediate, or high, a crucial step for further disease control [15]. While in our study the majority of patients were diagnosed with stages cT2c-cT4 (467 patients, 94%) and/or locally advanced disease (303 patients, 60%), many of them were diagnosed with a preoperative PSA level < 10 ng/mL (329 patients, 66%); at the same time, only 72 (14%) patients experienced actual BCR or showed positive lymph node status (80 patients, 16%) (Table 1). With such heterogenic phenotypes, many patients might not share the improvements in their disease management, being a vastly different subset of a particular EAU risk group. Indeed, they were characterized by separate clinical entities arising from a set of molecular properties culminating in PC of a higher invasiveness; even with initial good prognosis or despite their unfavorable characteristics, they were devoid of metastatic potential and thus will never progress beyond PSA—only recurrence. Therefore, the current stratification of the risk of BCR in clinical practice was deemed ineffective, either underscoring or overestimating such cases, especially regarding the recent debate on PSA’s usefulness and its accuracy in predicting the risk of BCR [12]. In support of the above, we showed that preoperative PSA values grouped as suggested in the EAU recommendations (<10 ng/mL, 10–20 ng/mL, ≥20 ng/mL) were not significant surrogates differentiating BCR prognosis (*p* = 0.72, Appendix A) in an independent manner, even though this was opposed to the available body of research (e.g., [13]). In turn, we established a cutpoint for the PSA level of 6.9 ng/mL that best differentiated the risk of BCR reaching an almost two-fold rise (Figure 1, Table 1) and was greater than the generally accepted level of 4 ng/mL utilized for PC screening [21].

An evolving body of research has attempted to provide a better framework for the prediction of BCR progression, going beyond the clinical characteristics per se, which, like PSA, are disputable, and enriching them with a molecular landscape reflecting the clinical behavior of PC. The above was also a rationale for our study, meriting an accurate stratification of PC patients deemed to be at an increased risk of a lethal course of the disease, which is, in fact, dictated by specific profiles of molecular markers of functional relevance in the mechanism of PC progression. For instance, the latest study by Pellegrino et al. employing the Decipher^®^ dataset showed that preoperative PSA blood levels poorly discriminated BCR, the development of metastasis, and PC-specific mortality (PCSM), with an area under the curve (AUC) of 50%, 51.5%, and 50.9%, respectively. At the same time, the study revealed the much higher accuracy of GGS, T score, and *KLK3* encoding PSA in predicting these events [40]. In parallel, we showed that even the optimized cutpoint for the preoperative PSA level was not superior in predicting BCR to the expression of functional markers involved in transcriptomic reprogramming that appeared to be significant in PC recurrence, as shown in previous studies [2,3]. In particular, we found that the expression cutpoints determined for *PGR*, *KLK3*, *CDH1*, and *MMP3* indicated a much higher risk (HR > 2) of BCR than preoperative PSA level, whereas the expression of *ESR1* and *VIM* showed a comparable association (Figure 1, Table 1). Moreover, it is worth noting that a higher expression of *KLK3* correlated with better clinical outcomes (HR = 0.423, 95% CI = 0.265–0.677, *p* = 0.0002, Figure 1, Table 1), and this finding was consistent with other research demonstrating that patients with positive lymph nodes, BCR, and metastatic disease, as well as those who died from PC, had significantly lower expression levels of *KLK3* [9,40]. The discrepancies between the serum PSA level and the tissue expression of KLK3 may be explained by the fact that during cancer development, there is an increased release of PSA into the blood [41]. Low expression levels of KLK3 are observed in poor-prognosis PC as a result of a loss of tissue differentiation [42]. On the other hand, some studies suggest that a higher expression of KLK3 may be associated with an indolent course of PC through KLK3 antiangiogenic activity [43,44]. It should also be noted that the alterations observed at the transcriptomic level are in fact a first line of response to signals received from the environmental milieu, though rarely translating into the expression of the corresponding proteins in a comparable ratio.

The androgen receptor is known for its role in the development and function of a normal prostate, but it is also an essential driver of prostate tumorigenesis. As a TF, it exerts specific effects in cells through the modulation of target genes followed by recruitment or crosstalk with other TFs and signaling pathways [45]. Since Huggins’ publication in 1941 on surgical castration in advanced PC with exceptional palliative effect, AR has become the main domain of interest in the treatment of locally advanced and metastatic PC [46]. Therefore, androgen deprivation therapy (ADT) achieved by pharmacological or surgical castration has become a therapeutic mainstay for such cases, as well as for adjuvant treatment in patients with BCR after primary local treatment [47]. As was shown, some of the tumors lost the expression of *AR*, but most retained it to reactivate the AR signaling axis under conditions of androgen refraction. By acquiring gain-of-function mutations, PC cells adapt to the low availability of androgens and de-differentiate into highly lethal mCRPC, which is in fact anticipated by BCR [48]. We previously reported that a higher expression of *AR* was associated with unfavorable disease-free survival (DFS) among PC patients, although the negative effect was diminished with progressing age, ultimately shifting in favor of signaling through *ESR1* [2]. Surprisingly, in the present study involving the same group of patients, the expression of *AR* was insignificant with regards to BCR (Figure 1, Table 1).

The overt role of AR in the prostate gland and PC puts it in the spotlight as a major research interest but, at the same time, should not detract from the relevance of other hormonal receptors such as estrogen or progesterone receptors (ER and PGR, respectively) in the evolution of PC [49,50,51]. The role of ER in PC progression has been debated since the local expression of estrogens and aromatase converting testosterone to E_2_ was detected in the malignant tissue of the prostate, thus showing that a single hormone may exert different effects depending on the receptor that it interacts with [52].

Bonkhoff et al. in 1999 revealed for the first time an increase in ESR1 expression correlating with metastatic PC lesions including lymph nodes [53] and, years later, demonstrated that progressing PC bypasses the androgenic cascade in favor of the re-emergence of the alternative ESR1 steroidal pathway [51], which resembled our findings.

These observations, pertaining to the clinical management of PC, underlaid, for instance, a clinical trial of selective ER modulators (SERMs) whose administration culminated in the prevention of further PC progression as a major clinical outcome in men [54], which had only been shown in vivo through the antagonism of ERα, not ERβ [55]. Moreover, compelling data demonstrate a significant contribution of estrogen signaling in the EMT, a propulsive factor in PC progression, invasiveness, and drug resistance [56]. Another piece of evidence supporting the deadly dualism of AR and ER in the trajectory of PC may also originate from their transcriptional targets, 80% of which, as we revealed, overlapped between the TFs, and which comprised a total of over twelve thousand genes, including major EMT effectors (Table 2). The above is in agreement with still-relevant suggestions to implement SERMs into the standard of care due to the fact that both androgens and estrogens may affect the local architecture of the PC cells through the direct governance of the EMT program, accounting for disease progression [57].

The molecular abundance of the pathways leading to BCR is a remarkable avenue in the improvement of risk stratification during the evolution of PC. Our previous investigations elucidated that the clinical manifestation of PC is mediated by the set of cellular effectors, i.e., the genes linked to the invasive properties of the tumor acting in concert [3] and, more importantly, in combination with clinical features such as GGS or age [2]. Even with the highest discriminative potential, none of the biomarkers could stratify the BCR risk individually. We subsequently attempted to extricate subgroups of PC patients with a markedly increased risk of BCR who shared a common profile of unfavorable alterations. Through the UpSet analysis, we identified 296 PRAD patients (60%) with simultaneous heightened *ESR1* and *MMP3* at the time of the primary treatment, which significantly correlated with worse BCR-free survival. Patients with that signature, termed in our study the stratum, showed significant disparities in regards to not only BCR (HR = 2.2, 95% CI = 1.3–3.6, *p* = 0.003; Figure 2, Table 2) but also other major clinical parameters. In particular, the carriers of the stratum were associated with worse clinical prognosis in comparison to similar clinical groups, i.e., regarding the lymph node status, GGS, T score, and EAU risk-associated stage group (Figure 2, Table 2). Importantly, stratum-positive patients classified with the same BCR risk group by EAU guidelines tended to be much more susceptible to PC progression or even had worse outcomes than those classified with a higher risk. More specifically, stratum-positive patients with Gleason 7 (4 + 3), considered to be at an intermediate risk of BCR by the EAU (5-year BCR-free rate = 48% (95% CI = 27–85%)), were at a greater risk of BCR than patients with Gleason 8 (5-year BCR-free rate = 78% (95% CI = 61–100%)) or, surprisingly, even patients with Gleason 9 (5-year BCR-free rate = 62% (95% CI = 44–87%)) (Figure 2, Table 2).

The involvement of the regional lymph nodes, classified as N1 in PC, corresponds to locally advanced disease and is regarded as a high-risk feature of BCR in addition to the PSA level or GGS. We managed to find that stratum-positive patients with simultaneous N0, therefore not considered at a high risk of recurrence, had an unexpectedly greater risk of a BCR event than the non-carriers of the stratum with invaded lymph nodes (N1). As a notable point, the combination of the *ESR1* and *MMP3* signature with N1 status put the patients at the highest risk of BCR (Figure 2, Table 2). In addition to the above, BCR appeared much more often at an early stage (up to 1-year post-RP) among the latter (40.3%), thus representing an overwhelming majority in comparison to the stratum-negative subgroup (12.5%). The above is all the more significant as it was demonstrated by Venclovas et al. that BCR occurring up to 1-year post-RP together with the involvement of the lymph nodes is an essential predictor of failure in PC management [58].

Pelvic lymphadenectomy (pLND) during RP failed in the improvement of oncological outcomes and survival. On the other hand, it provided crucial information regarding staging, and therefore prognosis, that could not be obtained by any other available procedure [26,59]. The decision to perform a pLND is made upon the preoperative estimation of the probability of lymph node invasion based on various nomograms (involving GGS, PSA level, and T score) that have shown comparable diagnostic accuracy (Briganti, Partin, MSKCC), while 94% of patients were shown to be correctly staged [60]. If the estimated nomogram risk exceeds 5% nodal involvement, it is an indication for pLND, whereas, in patients with a low-risk disease, this procedure is not performed during RP. Of special emphasis are thus our findings regarding the actual implications for PC patients, that those classified primarily with a low risk of progression should not undergo pLND, but if harboring the stratum, and thereby being exposed to a higher risk of BCR, they might benefit from a pLND.

As mentioned above, we did not observe differential BCR-free survival according to the EAU-recommended PSA level groups (Appendix A). Instead, we showed a comparable risk of BCR in stratum-negative patients that did not increase in different PSA-level groups (considered as presenting either a low, intermediate, or high risk of BCR), although this finding was limited by the low number of events in that group. On the other hand, patients harboring the stratum presented not only a higher frequency of the BCR events, but also an advancing risk corresponding to the PSA level, with an HR = 1.79 (95% CI = 0.92–3.49) for PSA < 10 ng/mL; HR = 2.23 (95% CI = 1–4.97) for PSA 10–20 ng/mL; and HR = 2.8 (95% CI = 1.16–6.94) for PSA > 20 ng/mL (Figure 2, Table 2). Integrating the available data, there was significant compliance in the area of the preoperative factors predicting BCR, which included PSA level, T and N scores, GGS, and risk classification by the EAU; however, as expected, GGS ≥ 8 and PSA ≥ 20 ng/mL were considered the strongest surrogates of BCR. By utilizing CA, we revealed that separate clinical features allocated PC patients to the high-risk group of BCR, including GGS ≥ 8, advanced T and N1 status, and PSA ≥ 20 ng/mL, which, consistent with the above, were considered indicators of an aggressive course of PC; in addition, these features corresponded to not only BCR itself but also the presence of the molecular *ESR1* and *MMP3* signature that was determined to anticipate unfavorable outcomes (Figure 3). Of the utmost relevance is the fact that the latter effects were reflected by the *ESR1*-driven molecular landscapes of the EMT effectors enhanced by *MMP3* activity. Specifically, the escalated invasive characteristics of the PC among stratum-positive patients likely emerged from the mesenchymal nature of the tumor cells, namely their capability to migrate, disseminate, and form metastases (Figure 4), thus predisposing them to a more aggressive course of PC, which we also observed in previous studies [2,3]. Remarkably, clinical features such as stage, T and N scores, PSA level, and GGS combined with the stratum corresponded to a heightened expression of mesenchymal-state markers such as *CTNNB1*, *FN1*, *MMP2*, *MMP3*, *MMP9*, *CDH2*, *ACTA2*, *SNAI2*, *VIM*, *ZEB1*, and *ZEB2*, accompanied by a decreased expression of epithelial markers such as *CDH1*, *LAMA1*, and *KRT18* (Figure 4). Additionally, with the elevated risk of BCR among the stratum-positive group, patients demonstrated an elevated expression of mesenchymal-state markers even when compared to clinical groups of worse prognosis, i.e., stratum-positive N0 vs. stratum-negative N1, stratum-positive localized PC vs. stratum-negative locally advanced PC, stratum-positive PSA < 10 ng/mL vs. stratum-negative PSA > 20 ng/mL, hence explaining the BCR-related failure in these groups (Figure 2, Table 2). The most essential discovery of this study, and at the same time the major conclusion that may be drawn from it regarding the potential clinical implications, was the case of the PC patients collectively classified in the EAU high-risk group for BCR by any of the clinical determinants, such as PSA ≧20 ng/mL, >T2c, N1, and GGS ≧8, who showed an even poorer prognosis of BCR, with an over two-fold increase in HR when the stratum was present (Figure 2, Table 2). This finding ultimately proved that the molecular drivers involved in the EMT program, which we determined in our study, strongly predisposed to a heightened risk of BCR and contributed to a more aggressive course of PC, irrespective of the clinical features that are taken into account in standard clinical settings. We speculate that these various trajectories of PC have their origin in the branching clonal evolution of the tumor cells, which manifests in multiple foci diverging in terms of genomic loading and histomorphological features such as the Gleason score, considered separate entities, albeit of a common progenitor (e.g., [61,62,63,64]). This phenomenon is currently explained through several types of clonal expansion (e.g., linear, branched, neutral, or macroevolution), which involve catastrophic rearrangements within the genome such as chromoplexy, chromothripsis, and kataegis [65]. However, more and more evidence indicates the significant contribution of the non-genomic programs culminating in cell plasticity, especially EMT [66], generating numerous phenotypic states of the same genome and enabling PC cells to escape from therapeutic pressure [67]. Moreover, the specific profiles of the EMT were shown to reflect particular Gleason patterns and thus, indirectly, a subpopulation of PC, arising from signaling via the *AR* and *ESR1* axes. As has been shown, during PC progression, the cells favor the latter, providing the advantage of an alternative source of steroidal signaling activated to retract from the androgens; hence, these patients become an AR-negative subpopulation, proving the initial potential for an aggressive course of PC and constituting a stratification niche preceding progression into CRPC or even mCRPC [68]. This may be associated with a switch in the relevance of the hormonal signaling occurring along with the ageing soma, which we previously demonstrated [2,3] and stands in agreement with the theory of age-induced carcinogenesis [65]. Finally, we were able to cross-validate the findings with the independent cohort of Rubicz et al. [31] showing a similar trend of an EMT shift towards the mesenchymal state with the heightened expression of *ESR1* and MMP3, notably marked according to the stage of PC (localized/locally advanced) and the GGS (Figure 5), which corresponded to our conclusions and proved the accuracy of the study.

In summary, BCR is a determinant of PC recurrence significantly increasing disease lethality through evolution to mCRPC. Therefore, the accurate prediction of the BCR risk is of the highest urgency in the management of PC patients. Our study provided evidence that specific molecular signatures of *ESR1* and *MMP3* predispose patients to experience unfavorable outcomes at the time of RP, even when analyzed in similar or different clinical groups that were primarily designed to stratify patients as having a low, intermediate, or high risk of BCR. We showed that patients collectively classified into the EAU high-risk group for BCR by all of the clinical determinants were at a much higher risk of BCR, with an over two-fold increase in HR when the expression of *ESR1* and *MMP3* was heightened. Furthermore, we demonstrated that the PSA level, despite its common use as a component of BCR stratification groups, had a poor correlation with BCR risk altogether. We additionally indicated that stratum-positive patients could benefit from pLND even when classified as being at a low risk of BCR by available clinical determinants. Nevertheless, as with any scientific study, our work was not without its limitations, such as the relatively small subgroups of patients sharing specific characteristics and the limited possibility of cross-validating the results, drawing a direct comparison, and implementing the results in clinical standards.

In overall, it is surprising that such fundamental factors as *ESR1* have not been investigated yet for their potential to prognosticate PC aggressiveness. Therefore, we hypothesize that *ESR1* has a major role in PC development and progression and could become a major focus of further investigations in PC theragnosis. Presumably, *ESR1* together with *MMP3* orchestrates the molecular landscape of clonal branching evolution, illustrating PC progression much better than the Gleason score, BCR, or PSA itself.

## 4. Materials and Methods

### 4.1. Data Retrieval

The cohort of prostate adenocarcinoma (PRAD; used interchangeably with PC throughout the study regarding the dataset employed) was retrieved from The Cancer Genome Atlas (TCGA) repository available through GDAC Firehose, containing expression transcriptomic data (RNAseqV2, level 3, RSEM normalized, data status of 28 January 2018) supported by the clinical characteristics of individuals. Records that lacked either expression or clinical information were excluded from the study, which eventually included a total of 497 patients.

### 4.2. BCR-Related Risk Group Classification

PRAD patients were stratified according to the risk of BCR based on the set of clinical features (preoperative PSA serum level, stage, GGS, and lymph node status) adapted from the EAU, European Association of Nuclear Medicine (EANM), European Society for Radiotherapy and Oncology (ESTRO), European Society of Urogenital Radiology (ESUR), and International Society of Geriatric Oncology (SIOG) Guidelines on Prostate Cancer (2020 Update) [14].

According to the TCGA enrollment form V4.7 102414, a BCR event among the PRAD patients was defined as two or more consecutively elevated PSA results greater than 0.2 ng/mL. The characteristics of the PRAD cohort are shown in Table 3.

### 4.3. Identification of the Downstream Targets of AR, ESR1, and ESR2 Transcription Factors (TFs) with a Special Focus on the Markers of the EMT

The downstream transcriptional targets of *AR*, *ESR1*, and *ESR2* were identified by the IDs P10275, P03372, and Q92731, respectively, in the Gene Transcription Regulation Database v21.12 (GTRD; http://gtrd.biouml.org/ (accessed on 28 November 2022)), comprising the most recent collection of ChIP-seq-proven TF-binding sites for human [69]. Table 4 presents a summary of the target genes identified.

The pivotal markers of the EMT were determined through the available literature (especially [28,29]) and the Molecular Signatures Database (MSigDB) v2022.1.Hs for humans [70]. The list of the chosen markers targeted by the studied TFs is presented in Table 5.

### 4.4. Determination of Optimal Cutpoints of Gene Expression Stratifying BCR-Free Survival

The clinical significance of the hormonal receptors such as *AR* (androgen receptor); *ESR1* and *ESR2* (estrogen receptor (ER) α and β, respectively); and *PGR* (progesterone receptor), as well as the other receptors relevant for prostate carcinogenesis genes, especially those involved in the mechanisms of recurrence (*NR3C1* (glucocorticoid receptor (GR)); *KLK2* (human kallikrein 2, hK2); *KLK3* (prostate-specific antigen, PSA); *FOLH1* (prostate-specific membrane antigen, PSMA); *CDH1* (E-cadherin); *VIM* (vimentin); *CTNNB1* (β-catenin); and *MMP2*, *MMP3*, and *MMP9* (matrix metalloproteinases 2, 3, and 9, respectively)), was investigated in relation to BCR-free survival by employing the EvaluateCutpoints shiny app and the cutp algorithm for optimal cutpoint determination to stratify patients into subgroups of differential prognosis [71]. In brief, the algorithm fit a Cox proportional hazards regression model to the binary (event) and continuous (survival time and biomarker value) covariates and based the statistics on the score derived. Then, for each cutpoint of a biomarker, the data were split into two subgroups of biomarker values below and above the cutpoint. Finally, the log-rank statistic was calculated for each unique element, returning the most significant factor. Notably, either “low” or “high” expression levels of the genes correlated with BCR prognosis were defined with the determined cutpoints; hence, they cannot be considered absolute values.

To determine the cutpoints and associate them with BCR prognosis, we used the following clinical variables as an input for the cutp algorithm: “patient.biochemical_recurrence” as an event (binary), combined “patient.days_to_last_followup” and “patient.days_to_first_biochemical_recurrence” as follow-up time, and the continuous expression of a specific gene as a biomarker. The follow-up time parameters were combined due to BCR events occurring before the end of the follow-up period in some patients.

### 4.5. Identification of Molecular Signature Altering BCR-Free Survival

To identify the subgroups of PRAD patients bearing a specific combination (hereinafter called “stratum”) of the expression profiles associated with unfavorable BCR prognosis determined according to the cutpoints, the UpSetR algorithm was applied (UpSetR R package, R version 4.2.2) [72]. For this purpose, PRAD patients were aggregated according to intersections of the dummy encoded expression of particular genes related to BCR-free survival outcomes. Subsequent BCR-free survival analysis was performed to compare the prognosis of patients with and without the identified stratum in combination with various clinical parameters.

### 4.6. Survival Analysis

Kaplan–Meier curves for BCR-free survival considering the stratum and clinical parameters were plotted by the survival and survminer R packages (R version 4.2.2) [73]. Hazard ratios (HRs) with 95% confidence intervals (95% CI) for BCR-free survival probability according to dichotomized covariates were evaluated with the coxph() R function employing the univariate Cox proportional hazards regression model (*p* < 0.05). Additionally, 2- and 5-year BCR-free survival probabilities with 95% CI were calculated, as well as the restricted mean (rmean), representing the expected value (i.e., a measure of average survival) of time to an event that corresponded to the area under the survival curve up to a specific time point with the standard error (se_rmean_). By analogy with the determination of the optimal cutpoint of gene expression stratifying BCR-free survival described in Section 4.4 of the Materials and Methods section, we used the input “patient.biochemical_recurrence” as an event (binary), as well as combined “patient.days_to_last_followup” and “patient.days_to_first_biochemical_recurrence” as follow-up time. This was modified by replacing the biomarker with the identified “stratum” status as a binary variable, a priori stratifying the BCR prognosis in combination with clinical features.

### 4.7. Correspondence Analysis (CA)

To study the topological association between the identified stratum (quartile-based subgroups of cross-validation from Rubicz et al.’s study [31]), BCR, and clinical parameters, we applied correspondence analysis (CA), which is the equivalent to principal component analysis (PCA) for contingency tables [74]. The analysis was performed with the FactoMineR and factoextra R packages (R version 4.2.2) [75].

### 4.8. Hierarchical Clustering by Stratum Combined with Clinical Parameters

The expression profiling of the major markers of the EMT among the subgroups of patients stratified according to the established molecular signature and additionally split by the clinical parameters was performed based on the hierarchical clustering of the median gene expression in particular subgroups with the Pearson distance metric and complete linkage method. The clustering was performed and visualized using the NMF and RColorBrewer R packages with the aheatmap() function (R version 4.2.2).

### 4.9. Validation of the Study

The findings presented herein were cross-validated with an independent study by Rubicz et al. evaluating the transcriptomic profiles of 503 localized prostatic tumors [31]. Due to the major limitations of the lack of information on biochemical recurrence status and follow-up time in the validation cohort, we were not able to determine and stratify patients according to cutpoints; therefore, we partially corroborated the primary results based on the TCGA data by splitting patients into quartiles of *ESR1* and *MMP3* expression and comparing the EMT expression profiles through hierarchical clustering as well as comparing the clinical characteristics therein. The characteristics of the validation cohort are shown in Table 6.

## Figures and Tables

**Figure 1 ijms-24-08399-f001:**
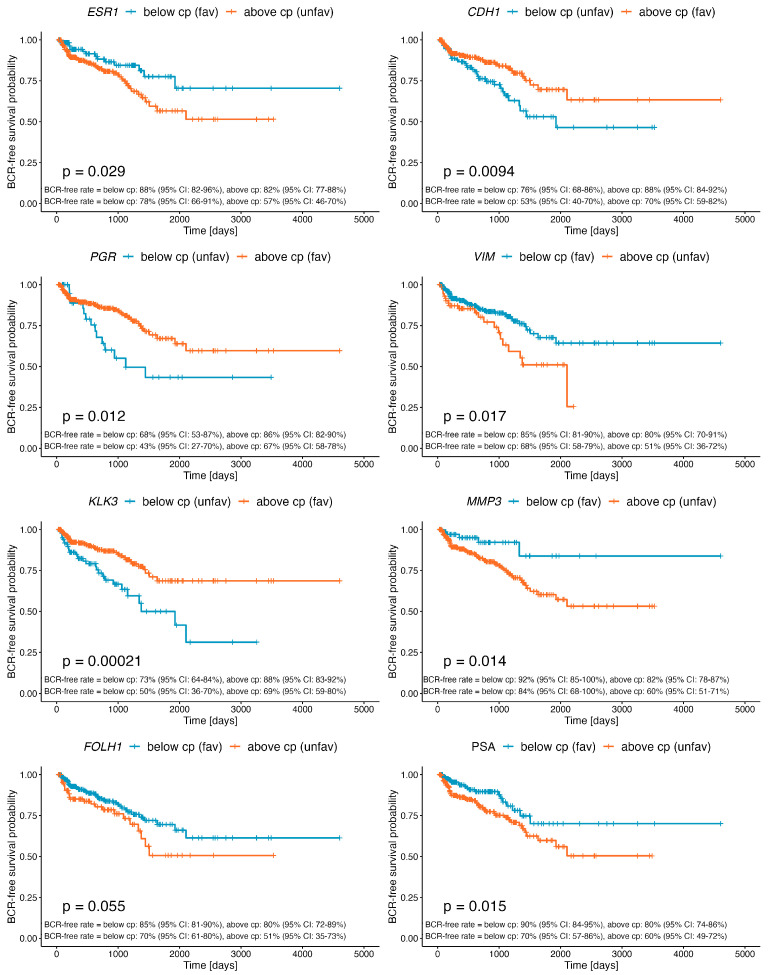
Kaplan–Meier curves demonstrating differential BCR-free survival of PRAD patients stratified by the determined cutpoint for the expression of *ESR1*, *PGR*, *KLK3*, *FOLH1*, *CDH1*, *VIM*, *MMP3*, and preoperative PSA level. Detailed statistics of the above analyses are presented in Table 1.

**Figure 2 ijms-24-08399-f002:**
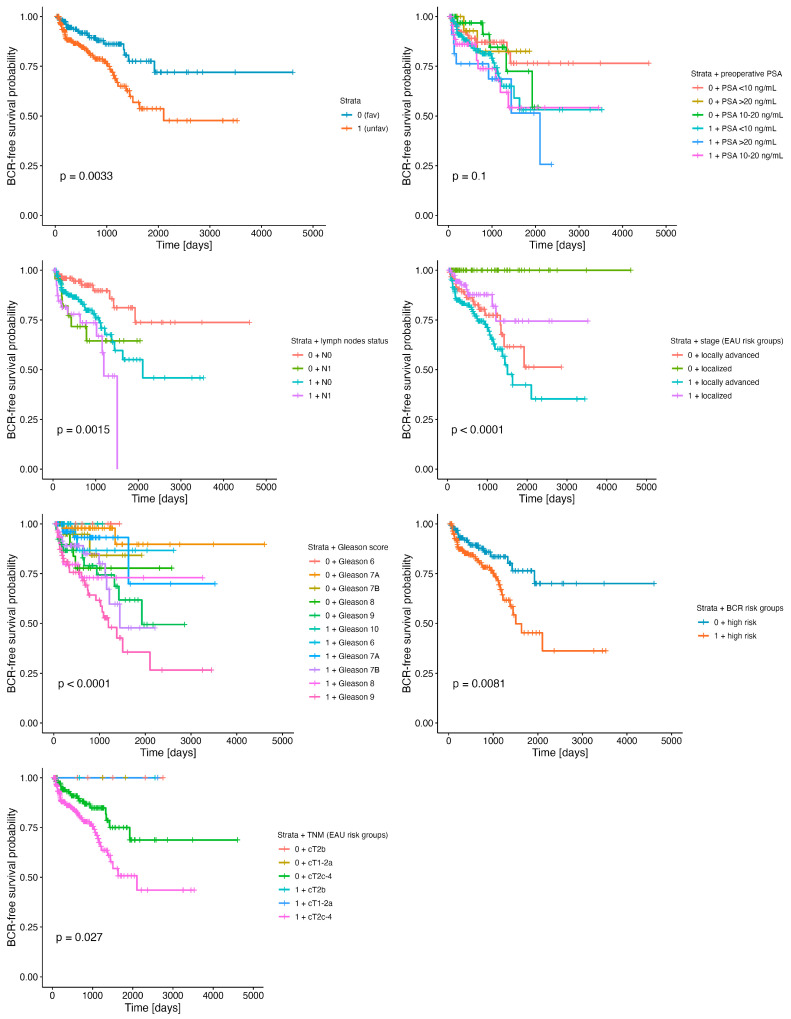
BCR-free survival differentiated by the stratum (unfavorable resultant profile of *ESR1* and *MMP3*) in combination with clinical parameters such as lymph node status, Gleason score, TNM, and stage (EAU risk groups for BCR), as well as preoperative PSA level groups. Detailed statistics of the above analyses are presented in Table 2.

**Figure 3 ijms-24-08399-f003:**
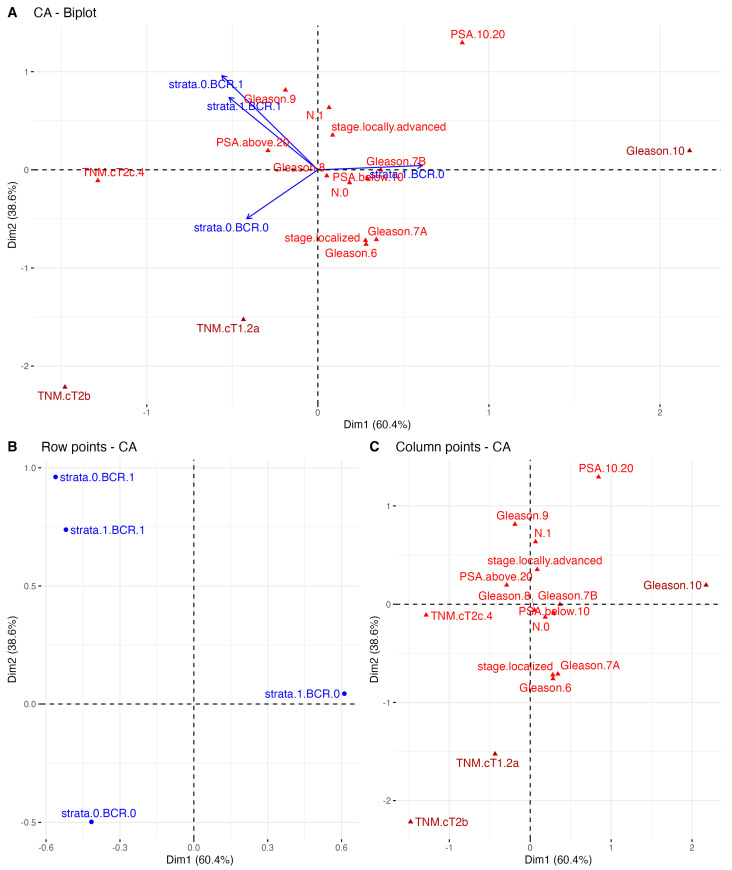
The correspondence analysis (CA) for TCGA PRAD of (**A**) both the stratum and clinical typologies; (**B**) only the row variables, i.e., stratum combined with the presence of BCR; and (**C**) the distribution of the column points only, i.e., clinical variables, partitioned along the two first dimensions of total variability reaching 99%. CA plot shows the distribution of the specific clinical features by different stratum- and BCR-related subgroups of patients. These features, which follow the arrows of the stratum and BCR status combined, were the most explanatory for the overall characteristics of each subgroup of PC patients.

**Figure 4 ijms-24-08399-f004:**
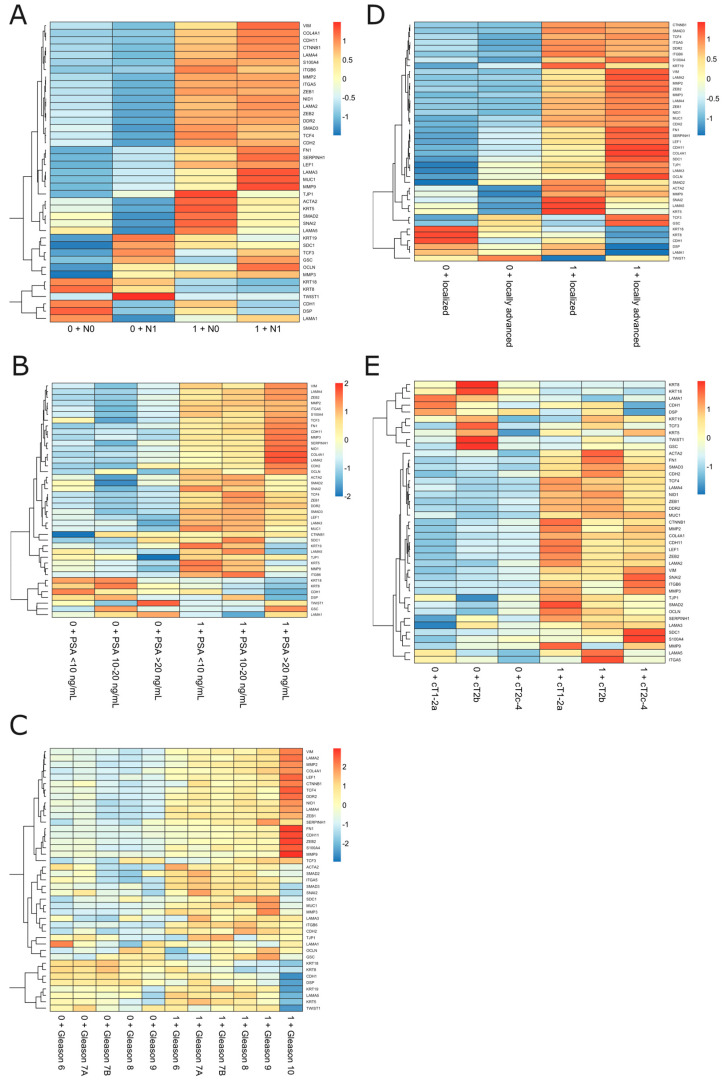
The EMT profiling among the subgroups of PRAD patients regarding stratum combined with clinical variables such as (**A**) lymph node status, (**B**) preoperative PSA level, (**C**) Gleason score, and (**D**,**E**) stage and TNM grouping according to the EAU BCR risk.

**Figure 5 ijms-24-08399-f005:**
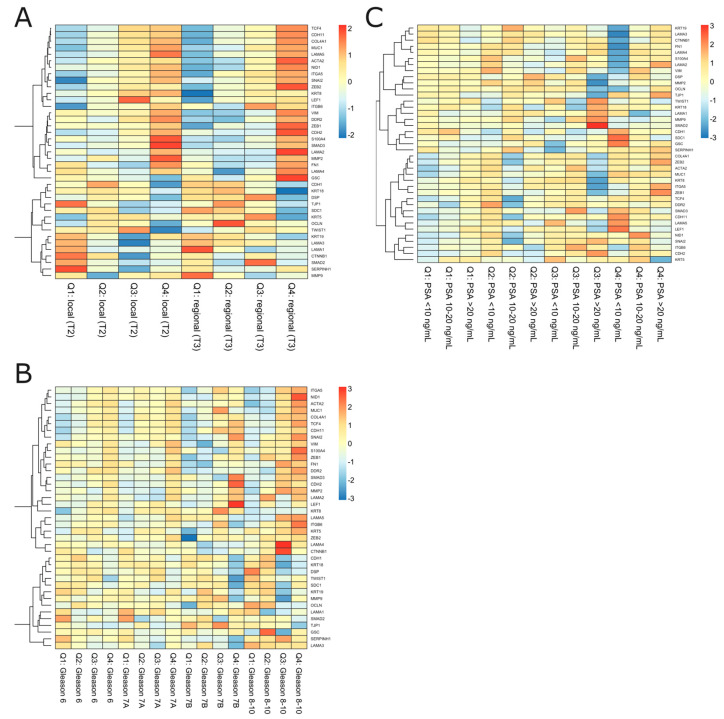
The EMT profiling among the subgroups of Rubicz et al. [31] cross-validation cohort regarding quartiles of *ESR1* and *MMP3* expression (Q1–Q4) combined with clinical variables such as (**A**) stage of localized prostate cancer, (**B**) Gleason score, and (**C**) preoperative PSA level.

**Table 1 ijms-24-08399-t001:** Detailed statistics of the BCR-free survival analysis according to the optimized expression cutpoints. The terms “low” and “high” define the expression level below or above the optimized cutpoint, respectively, and the associated BCR prognosis.

	Cutpoint	n < cp	n > cp	HR (95% CI)	*p*-Value	rmean (se_rmean_)	n_0_	n_events_	Endtime
*AR*	560.1	278	219	1.46 (0.917–2.32)	0.109				
*ESR1*	137.3	145	352	1.87 (1.06–3.31)	0.029				
Low (fav)						3551.66 (275.68)	122	15	4604
High (unfav)						2295.26 (154.35)	298	57	3524
*ESR2*	11	246	251	1.45 (0.903–2.33)	0.122				
*PGR*	46.91	61	436	0.466 (0.267–0.813)	0.006				
Low (unfav)						1905.99 (286.66)	44	15	3487
High (fav)						3180.63 (203.53)	376	57	4604
*NR3C1*	528.5	87	410	0.687 (0.406–1.16)	0.16				
*KLK2*	138,600	276	221	0.677 (0.418–1.1)	0.11				
*KLK3*	253,100	123	374	0.423 (0.265–0.677)	0.0002				
Low (unfav)						1742.94 (207.64)	104	30	3253
High (fav)						3442.65 (182.64)	316	42	4604
*FOLH1*	19,080	360	137	1.6 (0.986–2.58)	0.054				
Low (fav)						3231.79 (212.73)	308	46	4604
High (unfav)						2210.95 (224.19)	112	26	3524
*CDH1*	9554	151	346	0.545 (0.342–0.868)	0.009				
Low (unfav)						2119.8 (202.9)	118	32	3524
High (fav)						3315.46 (234.69)	302	40	4604
*VIM*	11,340	407	90	1.85 (1.11–3.07)	0.017				
Low (fav)						3298.58 (192.62)	345	51	4604
High (unfav)						1474.64 (120.12)	75	21	2211
*CTNNB1*	8689	321	176	1.39 (0.863–2.25)	0.172				
*MMP2*	3529	419	78	1.46 (0.834–2.54)	0.184				
*MMP3*	0.4285	91	405	2.93 (1.18–7.27)	0.015				
Low (unfav)						3999.75 (302.74)	76	5	4604
High (fav)						2341.89 (134.3)	344	67	3524
*MMP9*	238.6	294	203	1.43 (0.902–2.28)	0.126				
PSA	6.9	224	283	1.93 (1.15–3.25)	0.012				
Low (fav)						3500.86 (252.03)	178	20	4604
High (unfav)						2258.85 (155.49)	242	52	3487

**Table 2 ijms-24-08399-t002:** Detailed statistics of the BCR-free survival analysis according to the identified stratum combined with clinical parameters.

Stratum	Clinical Variable	n_0_	n_events_	rmean (se_rmean_)	Endtime	HR (95% CI)	*p*-Value	2-Year BCR-Free Rate (95% CI)	5-Year BCR-Free Rate (95% CI)
0		169	19	3602.05 (239.49)	4604	ref		89% (84–95%)	78% (67–90%)
1		251	53	2198.3 (167.05)	3524	2.2 (1.3–3.6)	0.003	81% (75–87%)	54% (42–68%)
	Lymph node status (pN)								
0	pN0	114	10	3707.84 (291.41)	4606	ref		93% (87–99%)	81% (69–96%)
	pN1	24	7	1443.27 (184.91)	2040	3.17 (1.2–8.3)	0.02	72% (55–94%)	64% (46–91%)
1	pN0	167	33	2202.1 (212.38)	3524	2.58 (1.27–5.2)	0.009	83% (76–90%)	55% (41–74%)
	pN1	44	13	1070.09 (99.5)	1506	4.65 (2.02–10.7)	<0.001	74% (60–91%)	-
	Gleason score								
0	6	18	0	1436 (-)	1436	-		-	-
	7A	56	2	4244.59 (269.05)	4604	ref		98% (94–100%)	90% (75–100%)
	7B	28	2	1714.53 (139.46)	1925	2.91 (0.41–20.7)	0.3	95% (85–100%)	84% (65–100%)
	8	26	4	2086.41 (217.07)	2576	5.77 (1.05–31.6)	0.043	78% (61–100%)	78% (61–100%)
	9	41	11	1933.1 (218.99)	2859	7.42 (1.64–33.5)	0.009	79% (66–95%)	62% (44–87%)
1	6	21	2	2336.27 (186.78)	2620	2.56 (0.36–18.2)	0.3	87% (71–100%)	87% (71–100%)
	7A	65	4	2863.35 (395.79)	3524	2.43 (0.44–13.3)	0.3	93% (86–100%)	70% (39–100%)
	7B	54	11	1555.68 (155.16)	2211	7.43 (1.65–33.6)	0.009	85% (74–97%)	48% (27–85%)
	8	33	7	2439.79 (274.56)	3253	9.05 (1.88–43.7)	0.006	73% (57–94%)	73% (57–94%)
	9	75	29	1582.5 (230.84)	3447	13.7 (3.27–57.4)	<0.001	69% (58–82%)	36% (21–61%)
	10	3	0	1065 (-)	1065	-		-	-
	TNM (EAU risk groups for BCR)								
0	cT1-2a	5	0	1815 (-)	1815	-		-	-
	cT2b	6	0	2753 (-)	2753	-		-	-
	cT2c-4	156	19	3489.94 (264.72)	4604	ref		88% (82–95%)	75% (63–89%)
1	cT1-2a	2	0	2620 (-)	2620	-		-	-
	cT2b	3	0	2553 (-)	2553	-		-	-
	cT2c-4	243	53	2110.36 (179.47)	3524	2.05 (1.21–3.46)	0.008	80% (74–86%)	51% (39–67%)
	Stage (EAU risk groups for BCR)								
0	Localized	70	0	4604 (-)	4604	-		-	-
	Locally advanced	97	19	1982.45 (175.46)	2859	ref		83% (74–92%)	62% (46–83%)
1	Localized	83	9	2817.88 (241.55)	3524	1.5 (0.87–2.57)	0.14	88% (79–97%)	74% (58–96%)
	Locally advanced	165	44	1870.44 (195.09)	3447	0.61 (0.28–1.36)	0.2	77% (70–85%)	42% (28–64%)
	Preoperative PSA groups								
0	<10 ng/mL	108	12	3707.59 (273.11)	4604	ref		87% (80–95%)	77% (63–94%)
	10–20 ng/mL	19	2	1628.96 (149.93)	2056	1.03 (0.36–2.93)	0.9	97% (91–100%)	72% (50–100%)
	>20 ng/mL	36	5	1739.72 (127.6)	1860	0.82 (0.18–3.69)	0.8	83% (63–100%)	83% (63–100%)
1	<10 ng/mL	171	31	2301.4 (212.54)	3524	1.79 (0.92–3.49)	0.088	83% (76–90%)	53% (38–75%)
	10–20 ng/mL	49	12	2206.48 (291.11)	2364	2.23 (1–4.97)	0.049	74% (60–91%)	54% (36–82%)
	>20 ng/mL	24	8	1495.88 (225.37)	3447	2.83 (1.16–6.94)	0.023	76% (60–91%)	51% (27–98%)

**Table 3 ijms-24-08399-t003:** Clinical summary of the PRAD cohort.

	n = 497
Age [years], median (range)	61 (41–78)
Preoperative PSA [ng/mL], median (range)	7.5 (0.7–107)
PSA (EAU risk groups for BCR)	
<10 ng/mL	329
10–20 ng/mL	99
>20 ng/mL	54
NA	15
TNM (EAU risk groups for BCR)	
cT1-2a	13
cT2b	10
cT2c-4	467
NA	7
Stage (EAU risk groups for BCR)	
Localized	187
Locally advanced	303
NA	7
Lymph node status (pN)	
N0	326
N1	80
NA	91
EAU risk groups for BCR	
Low risk	2
Intermediate risk	11
High risk	391
NA	93
Tumor status	
Tumor free	288
With tumor	54
NA	155
Vital status	
Alive	489
Dead	8
Laterality	
Bilateral	432
Left	19
Right	38
NA	13
Zone of origin	
Central zone	4
Overlapping/multiple zones	127
Peripheral zone	137
Transition zone	8
NA	221
Gleason score	
6	45
7A (3 + 4)	146
7B (4 + 3)	101
8	64
9	137
10	4
Primary pattern	
3	198
4	250
5	49
Secondary pattern	
3	152
4	235
5	110
BCR status	
No	351
Yes	78
NA	68

**Table 4 ijms-24-08399-t004:** Summary of target gene identification regulated by specific transcription factors via GTRD database. Venn diagram presents the overlaps between the targets of *AR*, *ESR1*, and *ESR2*.

	GTRD ID	No. of Mapped Target Genes
*AR*	P10275	14,390
*ESR1*	P03372	15,405
*ESR2*	Q92731	1057
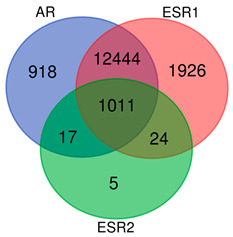		

**Table 5 ijms-24-08399-t005:** The most essential EMT-associated genes targeted by *AR*, *ESR1*, and *ESR2* based on [28,29,69].

	Gene Name	Marker of
*CDH1*	Cadherin 1, E-cadherin	Epithelial state
*COL4A1*	Collagen type IV α-1 chain	
*DSP*	Desmoplakin	
*KRT18*	Keratin 18	
*KRT19*	Keratin 19	
*KRT5*	Keratin 5	
*LAMA1*	Laminin subunit α-1	
*LAMA2*	Laminin subunit α-2	
*LAMA3*	Laminin subunit α-3	
*LAMA4*	Laminin subunit α-4	
*LAMA5*	Laminin subunit α-5	
*MUC1*	Mucin 1	
*NID1*	Nidogen 1	
*OCLN*	Occludin	
*TJP1*	Tight junction protein 1	
*ACTA2*	Actin 2, α 2	Mesenchymal state
*CDH11*	Cadherin 11	
*CDH2*	Cadherin 2, N-cadherin	
*CTNNB1*	Catenin β 1	
*DDR2*	Discoidin domain receptor tyrosine kinase 2	
*FN1*	Fibronectin 1	
*GSC*	Goosecoid homeobox	
*ITGA5*	Integrin subunit α 5	
*ITGB6*	Integrin subunit β 6	
*KRT8*	Keratin 8	
*LEF1*	Lymphoid enhancer binding factor 1	
*MMP2*	Matrix metalloproteinase 2, Gelatinase A	
*MMP3*	Matrix metalloproteinase 3, Stromelysin 1	
*MMP9*	Matrix metalloproteinase 9, Gelatinase B	
*S100A4*	S100 calcium binding protein A4	
*SDC1*	Syndecan 1	
*SERPINH1*	Serpin family H member 1	
*SMAD2*	SMAD family member 2	
*SMAD3*	SMAD family member 3	
*SNAI2*	Snail family transcriptional repressor 2	
*TCF3*	Transcription factor 3	
*TCF4*	Transcription factor 4	
*TWIST1*	Twist family bHLH transcription factor 1	
*VIM*	Vimentin	
*ZEB1*	Zinc finger E-box binding homeobox 1	
*ZEB2*	Zinc finger E-box binding homeobox 2	

**Table 6 ijms-24-08399-t006:** The clinical summary of Rubicz et al.’s cohort [31] used for the cross-validation of the primary findings.

	n = 503
Age group (years)	
35–49	67
50–54	90
55–59	123
60–64	145
65–69	53
70–74	25
Preoperative PSA for EAU BCR risk groups	
<10 ng/mL	369
10–20 ng/mL	72
>20 ng/mL	29
NA	33
Stage	
Local	338
Regional	165
Gleason score	
6	239
7A (3 + 4)	184
7B (4 + 3)	40
8–10	40

## Data Availability

Publicly available datasets were analyzed in this study. The data for PRAD can be found in the GDAC Firehose repository (https://gdac.broadinstitute.org/ (accessed on 10 October 2022)). The cross-validation data from Rubicz et al. [31] can be found in the Gene Expression Omnibus (accession number: GSE141551; https://www.ncbi.nlm.nih.gov/geo/ (accessed on 12 December 2022)).

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
