# Peer review of "The Transcriptomic Profiles of ESR1 and MMP3 Stratify the Risk of Biochemical Recurrence in Primary Prostate Cancer beyond Clinical Features"

_ijms, 2023, doi:10.3390/ijms24098399_

Round 1
Reviewer 1 Report
Brief Summary: The aim of the study by Olczak & Orzechowksa et al., determine molecular markers predictive of prostate cancer progression. The authors analyzed the PRAD-TCGA database using the cutp algorithm and identified gene signatures that associate with shorted biochemical recurrence (BCR)-free survival. They further describe that higher expression of the ESR1 and MMP3 genes correlates with worst clinical outcome in this patient dataset, suggesting an implication of epithelial-to-mesenchymal transition (EMT) in prostate cancer progression and recurrence. Interestingly, the predictive function of these molecular markers appeared to be independent of other clinical parameters, such as PSA levels, Gleason score or lymph node metastasis status. Overall, this is an interesting study describing an emerging prognostic role for the ESR1, MMP3 and EMT gene signatures for prostate cancer progression. However, this study feels to be purely descriptive and does not offer any functional analyses/input the role of identified genes/signatures to prostate cancer progression or recurrence. It is also unclear what additional information this study presents compared to the results previously published by the authors (PMID: 29206234).
Abstract:
· Overall, well-written and clear.
Introduction:
· The introduction provides an overview of prostate cancer epidemiology, risk assessment and treatment. However, some discrepancies are noted:
· Prostate cancer is not the fifth leading cause of death in men, but rather the second in mortality after lung cancer according to the latest cancer statistics (PMID: 36633525). The authors should correct this (it is correct in the Discussion section) [major].
· The authors fail to cite (here and in the Discussion section) critical literature on molecular biomarkers that can define clinical outcome such as NKX3.1, HOXB13 and BRCA2 (PMID: 22236224, 35490919, 33893149). Comprehensive description and discussion of literature is important. [major]
Materials and Methods:
· Clear and easy to follow;
· The authors should briefly describe the cutp algorithm, if space allows. [minor]
Results:
· Overall, the results are well-described in text and tables but, the figures are complicated and at time not very informative. [major]
· Figure 1 is not informative, p=values are missing in Figure 2, Figures 3 and 5 are not clear as to the data presented. [major]
· The additional validation cohort (Supplementary Figure 4) is better suited as a main figure to support the authors’ statements. [minor]
· The application and relevance of the CA analysis is unclear. The authors should better describe/explain why they applied these analyses. [major]
Discussion:
· The point on the clinical application of pLND for low EAU risk patients is very interesting and novel.
· The result that KLK3 expression (gene encoding for PSA) have an inverse correlation to outcome is interesting and requires further discussion. Since it is rather unexpected, the authors should also present and discuss literature that supports the opposite for a more comprehensive view. [minor]
· The adversative relationship between KLK3 and FOLH1 expression with regards to clinical outcome is not discussed. This is an important point to clarify. [major]
· The data is adequately discussed in most part, but the authors could note the limitations of their study. [minor]
Author Response
29th APRIL 2023
Dear Reviewer,
We appreciate the time and effort that you have dedicated to providing your helpful feedback on our manuscript. We thank you for allowing us to submit a revised draft of our manuscript titled ”The transcriptomic profiles of ESR1 and MMP3 stratify the risk of biochemical recurrence in primary prostate cancer beyond clinical features.” We have taken all your valuable comments into account and we hope that the provided amendments improved the manuscript accordingly to meet your requirements.
Brief Summary: The aim of the study by Olczak & Orzechowksa et al., determine molecular markers predictive of prostate cancer progression. The authors analyzed the PRAD-TCGA database using the cutp algorithm and identified gene signatures that associate with shorted biochemical recurrence (BCR)-free survival. They further describe that higher expression of the ESR1 and MMP3 genes correlates with worst clinical outcome in this patient dataset, suggesting an implication of epithelial-to-mesenchymal transition (EMT) in prostate cancer progression and recurrence. Interestingly, the predictive function of these molecular markers appeared to be independent of other clinical parameters, such as PSA levels, Gleason score or lymph node metastasis status. Overall, this is an interesting study describing an emerging prognostic role for the ESR1, MMP3 and EMT gene signatures for prostate cancer progression. However, this study feels to be purely descriptive and does not offer any functional analyses/input the role of identified genes/signatures to prostate cancer progression or recurrence. It is also unclear what additional information this study presents compared to the results previously published by the authors (PMID: 29206234).
Response:
In general, this study presents a holistic view of the molecular aspects of PC progression that are manifested in clinical characteristics observed among patients. It is a functional analysis of molecular determinants of worse prognosis and course of the disease than it would appear from the clinical features like EAU risk groups for instance. The selection of the considered genes is not random; we designed the study a priori oriented on the relevant genes to the major mechanism of PC progression and recurrence, i.e. epithelial-to-mesenchymal transition (EMT). In the Introduction part, we briefly described the molecular basis of the EMT program driving the progression of cancer and referred to our previous findings that its transcriptomic profiles directly reflect the Gleason score, this is to say an aggressiveness and invasive potential of PC cells. Although, that research did not go beyond the Gleason score to routinely classified subgroups of patients according to e.g. EAU guidelines and did not correlate the molecular profiles to any specific prognosis.
Regarding the methodology, we initially correlated the expression of the most clinically pivotal PC-associated genes of functional significance (major hormonal receptors and EMT effectors/markers, among others) with the biochemical recurrence that anticipates unfavorable outcomes. The subsequent analyses involved in turn the transcriptional targets of ESR1 (which significantly differentiated BCR-related prognosis in support of MMP3). We considered only those downstream effectors of ESR1 signaling that are participating in the EMT program underpinning progression, aggressiveness, and worse clinical outcomes among PC patients (shown in Table 5). Therefore, we believe we did offer a set of functional analyses, which results from the molecular function of the genes and gene sets explaining observed phenotypes and outcomes. Finally, we stated in lines 124-127 that we aimed “to determine the association of BCR with its molecular drivers, i.e. to identify functional biomarkers of pivotal relevance in EMT, with regards to clinical parameters such as PSA, GGS, and T score that assess the risk of BCR at the time of primary treatment and enable to stratify the potential to more aggressive course of PC.” Nevertheless, we modified slightly the corresponding paragraphs of the manuscript to emphasize all of the above.
Abstract:
- Overall, well-written and clear.
Introduction:
- The introduction provides an overview of prostate cancer epidemiology, risk assessment and treatment. However, some discrepancies are noted:
- Prostate cancer is not the fifth leading cause of death in men, but rather the second in mortality after lung cancer according to the latest cancer statistics (PMID: 36633525). The authors should correct this (it is correct in the Discussion section) [major].
- The authors fail to cite (here and in the Discussion section) critical literature on molecular biomarkers that can define clinical outcome such as NKX3.1, HOXB13 and BRCA2 (PMID: 22236224, 35490919, 33893149). Comprehensive description and discussion of literature is important. [major]
Response - Introduction
The PC statistics that we have quoted in our manuscript refer to worldwide epidemiology, namely Cancer statistics for the year 2020 as part of the latest International Agency for Research on Cancer (IARC) GLOBOCAN (PMID: 33818764). According to the above report PC in men with 1,414,000 new cases annually is second right after lung (1,435,000 new cases annually) and fifth malignant tumor causing death with 375,000 events annually after lung (1,188,000 deaths annually), liver (577,000 deaths annually), stomach (500,000 deaths annually) and esophagus cancer (374,000 deaths annually). Similar conclusions can be drawn from other publications (PMID: 34716119, PMID: 31068988, PMID: 34884434). Your concerns and discrepancies between the quoted statistics originate from different populations, i.e. we quoted worldwide epidemiology, while the publications you suggested are solely based on epidemiology among the population of the United States; therefore, we decided not to provide any changes in that matter.
We have added a paragraph regarding the importance of biomarkers you suggested providing the appropriate references. Although we kept the paragraph short as some kind of a highlight as they are aside from the major scope of our study. Moreover, the tumors arising as a result of germline mutations or variants are having a different etiology, which should not be associated with the molecular basis of sporadic carcinogenesis. We agree it should be mentioned, but we did not expand on them as a comprehensive review to avoid excessive and potentially misleading content of the article.
Materials and Methods:
- Clear and easy to follow;
- The authors should briefly describe the cutp algorithm, if space allows. [minor]
Response - Materials and Methods
We added a brief description of the cutp algorithm and the details how the analysis was performed.
Results:
- Overall, the results are well-described in text and tables but, the figures are complicated and at time not very informative. [major]
- Figure 1 is not informative, p=values are missing in Figure 2, Figures 3 and 5 are not clear as to the data presented. [major]
- The additional validation cohort (Supplementary Figure 4) is better suited as a main figure to support the authors’ statements. [minor]
- The application and relevance of the CA analysis is unclear. The authors should better describe/explain why they applied these analyses. [major]
Response - Results
We think that the general BCR-free survival of the TCGA cohort should be shown, although, as it is not a major finding of the study we moved it to the Supplementary Data (Supplementary Figure 1). We added global p-values to Figure 2 according to your suggestion, but the most important and detailed statistics, with p-values, are gathered in the corresponding Table 1. The description of Figure 5 was expanded to facilitate its understanding. This graph is a typical CA plot and is generated by the corresponding algorithm; therefore, we do not have any influence on the overall design. Besides, we justified more clearly CA analysis. Figure 3 as less important and showing the numbers only has been moved to Supplementary Data (Supplementary Figure 2). According to your suggestion, we also moved Supplementary Figure 4 to the body of the manuscript. We agree with your feeling that it is important to show the corroboration results.
Discussion:
- The point on the clinical application of pLND for low EAU risk patients is very interesting and novel.
- The result that KLK3 expression (gene encoding for PSA) have an inverse correlation to outcome is interesting and requires further discussion. Since it is rather unexpected, the authors should also present and discuss literature that supports the opposite for a more comprehensive view. [minor]
- The adversative relationship between KLK3 and FOLH1 expression with regards to clinical outcome is not discussed. This is an important point to clarify. [major]
- The data is adequately discussed in most part, but the authors could note the limitations of their study. [minor]
Response - Discussion
We agree that the inverse correlation between KLK3 expression and serum level of encoded PSA might seem unexpected. However, our observations are consistent and confirmed the other reports. Gene expression does not translate into protein at a comparable level, and more importantly, it would be very difficult to further compare it with serum levels of the protein released from cells. Thereby, we referred to our best knowledge to the most relevant and available literature delineating such findings.
We agree that the relationship between KLK3 and FOLH1 expression is of interest and should be clarified, although the FOLH1 was marginally correlated with the BCR-free survival, and thus was purposely omitted in further considerations. Due to excessive content and multiple aspects covered, in the Discussion, we mainly focused on these results that are clinically relevant to either BCR or PC progression and are delineated by the molecular background. We aimed to keep the major idea concise throughout the whole manuscript and stick to the major findings, even though we realize that some aside results are also very interesting and worth emphasis. We agree that the manuscript lacks clearly stated limitations of the study; therefore, we added an appropriate paragraph in the Discussion.
Sincerely,
Michał Olczak
II Clinic of Urology
Medical University of Lodz
mr.olczak@kopernik.lodz.pl

Reviewer 2 Report
The manuscript of MichaÅ‚ Olczak reports that ESR1 plays an important role in PC progression, thus future studies should be performed to clarify its role in PC progression. The authors perform their studies using the TCGA transcriptomic and clinical data of 497 individuals’ population with PC, stratified according to the risk of BCR by EAU-EANM-ESTRO-ESUR-SIOG.
In my opinion, the manuscript presents important new information necessary to better understand the progression of prostate cancer and how to prevent adverse events with suitable treatments.
However, some changes need to be made to increase the robustness of this manuscript
1- in figure 3 the font size should be increased to make it easier to read
2- same request for legend reported in figure 2 and 3.
3- please delete the MDPI information for chapter 5. It is not part of the manuscript
4- In the discussion, it should be emphasized that the population examined is small (497 patients) so that the reader understands that this is data to be validated with further studies
Author Response
29th APRIL 2023
Dear Reviewer,
We appreciate the time and effort that you have dedicated to providing your helpful feedback on our manuscript. We thank you for allowing us to submit a revised draft of our manuscript titled ”The transcriptomic profiles of ESR1 and MMP3 stratify the risk of biochemical recurrence in primary prostate cancer beyond clinical features.” We have taken all your valuable comments into account and we hope that the provided amendments improved the manuscript accordingly to meet your requirements.
Sincerely,
Michał Olczak
II Clinic of Urology
Medical University of Lodz
mr.olczak@kopernik.lodz.pl

Reviewer 3 Report
"While the authors acknowledge as ONLY limitation of their study “the lack of information on biochemical recurrence status and follow-up time” (lines 634-635), claiming the stratification of the risk for Biochemical Recurrence (BCR) based on the ONLY missing major information is speculative, and all their work appears as a complex fishing expedition.
The investigation is interesting overall, but the main message I got reading the manuscript is about an exploratory hypothesis which must be deeply and further investigated in a pre-defined cohort of patients. This should allow obtaining tumor tissue followed by biopsy for comprehensive transcriptomic and proteomic data (aka, ESR1 and MMP3 expression by IHC vs. gene expression). Furthermore, the soluble circulation ESR1 and MMP3 should be investigated followed by corelative studies between basal and follow-up status (for soluble ESR1 and MMP3) of these markers and BCR recurrence.
Changes in gene expression do not translate to changes in protein expression and their function, post translation modification, phosphorylation, signaling etc."
please check some word
Author Response
29th APRIL 2023
Dear Reviewer,
We appreciate the time and effort that you have dedicated to providing feedback on our manuscript.
We agree that the manuscript lacks a comprehensive description of the limitations of the study in general. These mentioned by you were pointed out as a reason for insufficient validation with an independent cohort only, while not considered as overall limitations. Therefore, we disagree with your further opinion that we stratified BCR-related prognosis without downright indispensable data like BCR event and follow-up time. To explain, it is to say, the primary analyses involved the stratification of BCR-free survival in correlation with the expression of clinically relevant genes. For that purpose, as stated in Materials and Methods (section 4.4), we employed the cutp algorithm to determine the optimal cutpoint to split patients into subgroups of favorable/unfavorable BCR-related prognosis. It is based on Cox proportional hazards regression model that requires input variables such as event (BCR) and follow-up time for any computations. Further models involving the identified "strata" were also based upon the BCR-free survival (classic Cox proportional hazards regression model, sections 4.5 & 4.6 of Materials and Methods). To emphasize that without leaving any doubts, we added a brief description of the cutp algorithm and clear information about the variables and settings included in the analysis. More importantly, without such data, we would not be able to provide any of the results shown in e.g. Table 2.
We admit that your further opinion is very true if we would indeed omit the crucial data in our analyses. Thanks to your opinion, it made it visible to us that perhaps the description is not clear enough for the reader, so we made some amendments to improve it. Our statement you mentioned “the lack of information on biochemical recurrence status and follow-up time” (lines 634-635) referred to the issues with full validation of the findings and this is the major reason why the study must be further investigated in a pre-defined cohort of patients; i.e. we validated the ESR1-MMP3-related transcriptomic profiles of the EMT program, but due to these lacks failed in validating the strata-related BCR-free survival.
We also agree that the changes in gene expression do not translate to protein levels. This study presents in turn a holistic view of the molecular aspects of PC progression that are manifested in clinical characteristics observed among patients. It is a functional analysis of molecular determinants of worse prognosis and course of the disease than it would appear from the clinical features like EAU risk groups for instance observed at the transcriptomic level as a first line of response to any alterations at the cellular system. Thus, we did not claim any translation into protein as we were able to explain the BCR-related phenotypes by a function of the concerned molecular background.
Sincerely,
Michał Olczak
II Clinic of Urology
Medical University of Lodz
mr.olczak@kopernik.lodz.pl

Round 2
Reviewer 1 Report
The authors have addressed the reviewer's concerns and the manuscript is now clearer. No further issues to address.
Reviewer 3 Report
I would like to thank the authors for their pertinent answers. I think that their manuscript is suitable for publication.
Thank you.